# Prevalence and patterns of substance use in West Africa: A systematic review and meta-analysis

**Godwin Omokhagbo Emmanuel[1], Folahanmi Tomiwa Akinsolu[2,3] \*, Olunike Rebecca Abodunrin[2,3,4], Oliver Chukwujekwu Ezechi[2,3]**

1 Mozuk Future Solutions, Nigeria, 2 Center for Reproduction and Population Health Studies, Nigerian Institute of Medical Research, Yaba, Lagos, Nigeria, 3 Department of Public Health, Faculty of Basic Medical and Health Sciences, Lead City University, Ibadan, Oyo, Nigeria, 4 Nanjing Medical University, Nanjing, China

\* Folahanmi.tomiwa@gmail.com, Akinsolu.folahanmi@lcu.edu.ng

## Abstract

### Introduction

Substance use is a growing public health concern in West Africa, contributing to significant morbidity, mortality, and socioeconomic challenges. Despite the increasing prevalence, comprehensive data on the patterns and factors influencing substance use in the region remain limited. This systematic review and meta-analysis aim to synthesize existing research on the prevalence and patterns of substance use in West Africa, providing critical insights for developing targeted interventions and policies.

### Methodology

This study followed the Preferred Reporting Items for Systematic Reviews and Meta-Analyses [PRISMA] guidelines. A systematic search was performed across four major databases [PubMed, Web of Science, CINAHL, and Scopus] from January 2000 to June 2024. A total of 22 studies involving 43,145 participants met the inclusion criteria. Multiple reviewers performed data extraction and quality assessment independently, and a random-effects meta-analysis was used to estimate the pooled prevalence of various substances. Sensitivity analysis was conducted using a leave-one-out approach to evaluate the influence of individual studies on the overall prevalence.

### Results

The meta-analysis revealed the pooled prevalence rates of alcohol [44%], cannabis [6%], tramadol [30%], codeine [11%], and kolanut [39%]. The analysis identified high heterogeneity across studies [$I^2$ = 98–99%], reflecting diverse substance use patterns and influencing factors, including peer influence, availability of substances, socioeconomic conditions, and demographic characteristics. Sensitivity analysis indicated that no single study significantly impacted the overall prevalence estimates, confirming the robustness of the findings.

**Data Availability Statement:** All datasets generated and analyzed, including the search strategy, list of included and excluded studies, data extracted, analysis plans, and quality assessment,

are available in the article and as supplementary files.

**Funding:** The authors received no specific funding for this work.

**Competing interests:** The authors have declared that no competing interests exist.

## Conclusion

Substance use in West Africa is widespread and influenced by complex factors. The high prevalence of alcohol and tramadol use highlights the urgent need for targeted public health interventions, including stricter regulatory frameworks, community-based prevention programs, and comprehensive public education campaigns. This study provides a critical foundation for developing effective strategies to mitigate the escalating substance use crisis in the region.

## Introduction

Globally, substance use is a major contributor to morbidity and mortality [1] and a significant public health concern, contributing to a myriad of social, economic, and health-related challenges. According to the World Drug Report, 2021 [2], approximately 275 million people used drugs worldwide in the preceding year, with another 36 million persons diagnosed with substance use disorders globally [2, 3]. In West Africa, the prevalence and patterns of substance use have garnered increasing attention due to the profound impact on public health, particularly among vulnerable populations [3]. The region is characterized by diverse socio-cultural dynamics, which influence drug use behaviors and access to prevention and treatment services [3, 4]. Understanding these patterns is essential for developing effective interventions tailored to the unique context of West Africa.

Substance use generally refers to the chronic consumption of substances, including alcohol and psychoactive drugs, in quantities or through methods that pose harm to the user or others [5–7]. This behavior often leads to varying levels of intoxication, which can impair judgment, attention, and perception [3, 5]. The use of alcohol, illicit drugs, and the misuse of prescription medications has been linked to significant health risks, reduced productivity, and a substantial socioeconomic burden on both families and society [1, 8]. Globally, the prevalence of substance use and substance use disorders is rising rapidly, accompanied by an increase in related morbidity and mortality [1, 9, 10]. In Africa, the consumption of illicit substances, such as cannabis [which remains the most widely used drug in the region, with a prevalence ranging from 5.2% to 13.5% in West and Central Africa], amphetamine-type stimulants, and benzodiazepines, is also increasing at an alarming rate [11].

Over the past decade, Africa has transitioned from being primarily a transit zone for illicit drugs, linking Latin America and Europe, to becoming a significant consumer and destination for these substances [3]. In Nigeria, for example, a recent national survey estimated that over 14.3% of the adult population has used a psychoactive substance, with a higher prevalence among males and young adults [8, 12]. Similarly, in Ghana, cannabis use among young people is a growing concern, with studies indicating that peer pressure and curiosity are major drivers of drug initiation [13]. The rising trend of opioid abuse, particularly tramadol, in countries like Sierra Leone and Guinea has also raised alarm, with these substances being linked to increased morbidity and mortality [10]. This shift is believed to have contributed to the rapid escalation of the substance use epidemic, particularly in urban centers across the continent.

The consequences of substance use in West Africa are far-reaching, impacting not only individual health but also community well-being and national development [3, 14]. Substance use has been associated with an increase in infectious diseases, such as HIV/AIDS and hepatitis, due to risky behaviors like needle sharing [11, 15]. Additionally, the burden of mental

health disorders linked to substance use is substantial, with many individuals suffering from co-occurring conditions such as depression, anxiety, and psychosis [11]. Furthermore, substance use contributes to social instability, crime, and violence, undermining efforts to achieve sustainable development in the region [16].

Despite the growing recognition of substance use as a critical public health issue, the response in West Africa has been hampered by several challenges [3, 9]. These include limited access to treatment services, inadequate funding for prevention programs, and a lack of comprehensive data on drug use patterns [17]. Most countries in the region lack robust surveillance systems to monitor drug use trends, which hinders the development of targeted interventions [4]. Moreover, stigma and discrimination against individuals who use drugs remain pervasive, further complicating efforts to provide care and support to those in need [18].

The primary objective of this systematic review and meta-analysis is to provide a comprehensive overview of the prevalence and patterns of substance use in West Africa. By synthesizing the available evidence, we aim to identify the types of substances commonly used, the routes of administration, and the health and social consequences associated with substance use in the region. Additionally, this review will pinpoint existing knowledge gaps and suggest priority areas for future research and policy development. Understanding the epidemiology of substance use in West Africa is vital for informing public health strategies tailored to the specific needs of the region.

## Methodology

### Protocol and registration

In conducting this systematic review, we adhered to the recommendations from the Preferred Reporting Items for Systematic Reviews and Meta-Analyses (PRISMA) statement [19]. A 27-item PRISMA checklist is available as an additional file to this protocol in S1 File. Our protocol was registered in the International Prospective Register of Systematic Reviews (PROSPERO): CRD42024563715.

### Search strategy

The search strategy was designed to identify studies reporting the prevalence and pattern of substance use in West Africa by systematically searching literature databases from January 2000 –June 2024. The four databases (PubMed, Web of Science, CINALH, and Scopus), Google Scholar, and Citation searching. Detailed search strategies for each database are reported in S2 File. The bibliographies of relevant reviews and eligible studies were also examined for additional sources. Databases containing conference proceedings, congress's annals, university theses, and experts were also consulted.

### Study selection

Following the initial search, all articles were loaded onto Rayyan Software (https://www.rayyan.ai/), where initial duplicate screening and removal were done [20]. After duplicate removal, the abstract and titles of retrieved articles were independently screened by two authors (FTA and ORA) based on pre-determined eligibility criteria. A second screening of full-text articles was also done independently by two authors (FTA and ORA). Disagreements during each screening stage were resolved through discussion and consensus by one of the authors (GOE). The list of excluded studies, along with the reasons for their exclusion during the full-text screening, is presented in the S3 File. Also, the list of excluded studies with their reasons excluded during the title and abstract screening is presented in S4 File.

## Studies eligibility criteria

The eligibility criteria for the studies included in this systematic review on the prevalence and patterns of substance use in West Africa were carefully defined to ensure the selection of relevant and high-quality research. Table 1 shows the criteria for inclusion and exclusion parameters based on population, outcomes, and study design (POS framework).

## Data extraction

Two independent authors (FTA and ORA) used a pretested data extraction form prepared in Microsoft Excel to extract details of articles that met the inclusion criteria independently. The information was the author's name, year of article publication, study design (cross-sectional, cohort-based), study location, and study setting were recorded. Information on the study participants (sample size and age range) was extracted. The prevalence of substance use in each study was also extracted. Discrepancies among reviewers during the extraction process were resolved by a third author (GOE).

See the details of the Extraction in S5 File.

## Data synthesis

The data synthesis for this systematic review on the prevalence and patterns of substance use in West Africa was conducted using a combination of narrative and quantitative approaches tailored to address the heterogeneity of the included studies.

A standardized data extraction form was used to gather detailed information from each included study. This information included study characteristics (author(s), publication year, country), study design, sample size, and demographic details of participants (age, gender, socio-economic status). Specific details on substance use, including types of substance use, levels of use, and patterns of use, were also extracted. These details were tabulated in a structured Excel to facilitate systematic comparison and analysis (See Table 2).

The narrative synthesis involved summarizing the findings of each study, focusing on the prevalence rates, types of substances commonly abused, demographic characteristics of substance users, and any reported patterns or trends in substance use.

The context and settings of the studies were considered to understand the socio-economic and cultural factors influencing substance use in West Africa. This qualitative analysis provided a comprehensive overview of the region's current state of substance use.

## Statistical method

To estimate the overall prevalence of substance, use across West Africa, we conducted a comprehensive meta-analysis that included studies reporting sample sizes and prevalence data for specific substances, provided that at least three studies were available for each substance. Data extraction involved carefully gathering information on sample sizes, prevalence rates, and

**Table 1. POS framework.**

| Population | • Studies included individuals of any age groups residing in West African countries. |
|---|---|
| Outcomes | • Studies reporting on the prevalence rates of substance use.<br>• Research identifying the types of drugs commonly abused.<br>• Studies providing demographic data, including age, gender, and socio-economic status of individuals with substance use disorders.<br>• Research documenting patterns of drug use, including polydrug use and trends over time. |
| Study design | • Observational studies, including cross-sectional, cohort, and case-control studies. |

**Table 2. Characteristics of included studies.**

| S/No | Author's Name/Year | Country | Study Type | Sample Size | Study Populations | Mean Age (Years) | Types of substances used | Factors influencing abuse | Route of substance use administration | Information on Study Tools | Quality Assessment |
|---|---|---|---|---|---|---|---|---|---|---|---|
| 1. | Abdulkarim et al., (2005) [22] | Nigeria | Cross-sectional study | 1200 | Students | 15.9 years | Kolanut, coffee, alcohol, sniffing agents, amphetamine, ephedrine, cigarette, heroin, cocaine, and cannabis | Peer influence, role model influence, religiosity, family supervision | • Kolanut and coffee are consumed orally. • Alcohol consumed orally. • Inhaled sniffing agents • Cigarettes: Smoking • Cannabis, Cocaine, Heroin: Smoked or inhaled. | Not Available | Low |
| 2. | Adebowale & James, (2018) [23] | Nigeria | Cross-sectional study | 395 | Pregnant women attending an antenatal clinic | 23.5 years | Alcohol and sedatives | - | • Alcohol: Ingested orally (drinking) • Sedatives: Taken orally or injected | • ASSIST developed by WHO was used. • Internal consistency was over 0.80 for most domains. It has also been validated for use in Nigeria, and its diagnostic accuracy was >95%. • SRQ-20 was used which was developed as part of a collaborative study coordinated by the WHO on strategies for extending mental health care. | Low |
| 3. | Aigbogun et al., (2024) [24] | Nigeria | Cross-sectional study | 520 | IDPs living in three camps located in Maiduguri, Borno state of Nigeria | - | Kolanut, sleeping pills, bitter kola, anabolic steroids, marijuana, codeine-containing syrups, cocaine, shoe polish, green tea, pit toilet gas, tramadol, glue, benzhexol, lizard dung. | Availability of substance, influence from others, having a disease condition | • Kolanut: Chewed or ingested orally. • Sleeping pills: Take them orally in pill form. • Bitter kola: Chewed or ingested orally. • Anabolic steroids: Injected or taken orally. • Marijuana: Smoked • Codeine-containing syrups: Ingested orally. • Cocaine: Snorted, smoked, injected • Shoe polish: Inhaled (huffing). • Green tea: Brewed and ingested as a beverage. • Pit toilet gas: Inhaled. • Tramadol: Taken orally. • Glue: Inhaled (huffing). • Benzhexol: Taken orally. • Lizard dung: Ingested orally." | • DUDIT was used wherein modifications were made by incorporating substances that were easily accessible at the study location into the existing list of substances available in the original version of DUDIT. | Low |

*(Continued)*

**Table 2.** (Continued)

| S/No | Author's Name/Year | Country | Study Type | Sample Size | Study Populations | Mean Age (Years) | Types of substances used | Factors influencing abuse | Route of substance use administration | Information on Study Tools | Quality Assessment |
|---|---|---|---|---|---|---|---|---|---|---|---|
| 4. | Aluh et al., (2023) [25] | Nigeria | Cross-sectional study | 520 | IDPs in camps located in Borno State, Nigeria | - | Kolanut, sleeping pills, bitter kola, anabolic steroids, marijuana, codeine-containing syrups, cocaine, shoe polish, green tea, pit toilet gas, tramadol, glue, benzhexol, lizard dung | Education, marital status, employment, and number of substances used | • Kolanut: Chewed or ingested orally. • Sleeping pills: Take them orally in pill form. • Bitter kola: Chewed or ingested orally. • Anabolic steroids: Injected or taken orally. • Marijuana: Smoked • Codeine-containing syrups: Ingested orally. • Cocaine: Snorted, smoked, injected • Shoe polish: Inhaled (huffing). • Green tea: Brewed and ingested as a beverage. • Pit toilet gas: Inhaled. • Tramadol: Taken orally. • Glue: Inhaled (huffing). • Benzhexol: Taken orally. • Lizard dung: Ingested orally. | • DUDIT was adapted to suit the study population. The drug list was changed to include the substances available in the study location. | Low |
| 5. | Amoah et al., (2022) [26] | Ghana | Cross-sectional study | 1929 | Men and women from Ghana | - | Aphrodisiacs and tramadol | Individual characteristics and behaviors, interpersonal factors, community norms and practices, institutional and public policy factors | • Aphrodisiacs and Tramadol: Oral consumption | Not available | Low |
| 6. | Anyanwu et al., (2017) [27] | Nigeria | Cross-sectional study | 620 | Students | 16.6 years | Alcohol, nicotine, cigarettes, cocaine and cannabis | Age group, gender, and socioeconomic class | • Alcohol: Oral consumption • Nicotine (in the form of kola nut and coffee), cigarettes, cocaine, and cannabis: Inhalation | Not available | Low |

(Continued)

**Table 2.** (Continued)

| S/No | Author's Name/Year | Country | Study Type | Sample Size | Study Populations | Mean Age (Years) | Types of substances used | Factors influencing abuse | Route of substance use administration | Information on Study Tools | Quality Assessment |
|---|---|---|---|---|---|---|---|---|---|---|---|
| 7. | Asante & Atorkey, (2023) [28] | Benin, Ghana | Cross-sectional study | 15,553 | Adolescents | - | Cannabis and amphetamine | Demographic characteristics (age and male gender), mental health factors (suicide ideation and attempt), lifestyle factors (cigarette smoking, past-month alcohol use, lifetime drunkenness, leisure-time sedentary behavior), school-level factors (truancy and bullying victimization), social support at school, parental monitoring, and parental tobacco use | • Cannabis: Inhalation Amphetamine: Intranasal (snorting) | Not available | Low |
| 8. | Bio-sya et al., (2022) [34] | Nigeria | Cross-sectional study | 627 | Students | 17 years | Alcohol, stimulants, tobacco, tramadol, fentanyl, THC, K2, Benzodiazepines, alcohol, methamphetamine, cotinine | Age | • Alcohol: Ingested orally (drinking). • Stimulants: Taken orally, snorted, or injected. • Tobacco: Smoked. • Tramadol: Taken orally. • Fentanyl: Taken orally, injected, or absorbed through skin patches. • THC (Tetrahydrocannabinol): Smoked, vaporized, or ingested orally (edibles). • K2 (Synthetic cannabinoids): Smoked or ingested orally. • BZDs (Benzodiazepines): Taken orally or injected. • Methamphetamine: Taken orally, snorted, smoked, or injected. | ASSIST developed by WHO was used. | Low |
| 9. | Danso & Anto, (2021) [35] | Ghana | Cross-sectional study | 458 | Male Commercial drivers and assistants | 42 years | Tramadol | Lack of parental control and peer influence | • Oral intake | ASSIST developed by WHO was used. | Low |

(Continued)

Table 2. (Continued)

| S/No | Author's Name/Year | Country | Study Type | Sample Size | Study Populations | Mean Age (Years) | Types of substances used | Factors influencing abuse | Route of substance use administration | Information on Study Tools | Quality Assessment |
|---|---|---|---|---|---|---|---|---|---|---|---|
| 10. | Forson et al., (2020) [36] | Ghana | Cross-sectional study | 171 | Trauma patients and non-trauma patients with altered mental statuses | - | Marijuana, opiates, oxycodone, cocaine, benzodiazepines, methamphetamines | - | •Oral intake | Not Available | Moderate |
| 11. | Gureje et al., (2007) [37] | Nigeria | Cross-sectional study | 6752 | Non-institutionalized adults | 44 years | Alcohol, tobacco, cannabis, sedatives, stimulants | Gender, age, religion, and income | •Alcohol: Consumed as beverages. •Tobacco: Smoked in cigarettes, cigars, or pipes. •Cannabis: Smoked. •Sedatives and Stimulants: Often taken orally as pills. | WHO CIDI was used for Diagnostic assessment. | Low |
| 12. | Idowu et al., (2018) [38] | Nigeria | Cross-sectional study | 249 | Secondary school students in Oyo state | 16.3 years | Alcohol, marijuana, codeine, tramadol, cough syrup | Peer pressure to be happy, just to have fun, to be more intelligent, to be more active | •Alcohol, Tramadol, Codeine, Cough syrup: Oral consumption Marijuana: Inhalation | Self-administered questionnaire developed by reviewing previous studies. | Low |
| 13. | Kyei-Gyamfi et al., (2024) [13] | Nigeria | Cross-sectional study | 4144 | Children and adolescents | 12.5 years | Alcohol, cigarettes, tramadol, marijuana, codeine, cocaine, heroin | - | •Alcohol: Ingested orally (drinking). •Cigarettes: Smoked. •Tramadol: Taken orally. •Marijuana: Smoked, vaporized, or ingested orally (edibles). •Codeine: Taken orally or as a syrup. •Cocaine: Snorted, smoked, injected. •Heroin: Injected, snorted, or smoked. | Not Available | Low |
| 14. | Lasebikan & Ijomanta, (2018) [39] | Nigeria | Cross-sectional study | 223 | Soldiers residing in a military community | - | Cannabis | Past disciplinary action in the workplace | •Cannabis: Smoked. | WHO CIDI was used for Diagnostic assessment. | Low |
| 15. | Makanjuola et al., (2010) [40] | Nigeria | Cross-sectional study | 2,143 | Nigerian general population | 39 years | Alcohol, tobacco, cannabis, cocaine, sedatives, stimulants, analgesics | Internalizing and externalizing disorders | •Alcohol: Ingested orally (drinking). •Tobacco: Smoked •Cannabis: Smoked •Cocaine: Snorted, smoked, injected. •Sedatives: Taken orally or injected. •Stimulants: Taken orally, snorted, or injected. •Analgesics: Taken orally or injected | •WHO CIDI was used for Diagnostic assessment. •Statistical manual of mental disorders of the American Psychiatric Association–Fourth Revision (DSM-IV). | Low |

(Continued)

**Table 2.** (Continued)

| S/ No | Author's Name/Year | Country | Study Type | Sample Size | Study Populations | Mean Age (Years) | Types of substances used | Factors influencing abuse | Route of substance use administration | Information on Study Tools | Quality Assessment |
|---|---|---|---|---|---|---|---|---|---|---|---|
| 16. | Mirian Aguocha & Nwefoh, (2021) [41] | Nigeria | Cross-sectional study | 763 | Students | 24 years | Alcohol, sedatives, cigarettes, tramadol, cannabis, codeine, cocaine, solvents, amphetamine, heroin | Age, religion, place of residence, family, and peer use of substances | • Alcohol: Ingested orally (drinking)<br>• Sedatives: Taken orally in pill form or injected<br>• Cigarettes: Smoked<br>• Tramadol: Taken orally<br>• Cannabis: Smoked, vaporized, or ingested orally<br>• Codeine: Taken orally as a syrup<br>• Cocaine: Snorted, smoked, injected<br>• Solvents: Inhaled (huffing).<br>• Amphetamine: Taken orally, snorted, or injected.<br>• Heroin: Injected, snorted, or smoked. | WHO CIDI was used for Diagnostic assessment. | Low |
| 17. | Obadeji et al., (2020) [42] | Nigeria | Cross-sectional study | 682 | Students | 15.75 years | Alcohol, tobacco, tramadol, cannabis, codeine, sedatives | Absence of parental care and parental substance use causing psychological distress and substance use risk | • Alcohol: Ingested orally (drinking).<br>• Tobacco: Smoked<br>• Tramadol: Taken orally<br>• Cannabis: Smoked<br>• Codeine: Taken orally as a syrup.<br>• Sedatives: Taken orally or injected. | • Substance use was assessed with the DSM-5 Level 2—Substance Use for adolescents, an adapted version of the NIDA-Modified ASSIST | Low |
| 18. | Odukoya et al., (2018) [29] | Nigeria | Cross-sectional study | 437 | School adolescents | 15.3 years | Alcohol, cigarette smoking, and marijuana | Parental monitoring practices, negotiated unsupervised time | • Alcohol: Oral consumption Cigarette<br>• Marijuana: Inhalation | • Substance use was measured using a modified version of the YRBSS questionnaire.<br>• The questionnaire was pre-tested and had a Cronbach's alpha coefficient of 0.84. | Low |

(*Continued*)

**Table 2.** (Continued)

| S/No | Author's Name/Year | Country | Study Type | Sample Size | Study Populations | Mean Age (Years) | Types of substances used | Factors influencing abuse | Route of substance use administration | Information on Study Tools | Quality Assessment |
|---|---|---|---|---|---|---|---|---|---|---|---|
| 19. | Olanrewaju et al., (2022) [30] | Nigeria | Cross-sectional study | 400 | Students | 19.5 years | Alcohol, cigarettes, codeine-containing syrup, tramadol, and cannabis | Emotional problems are predominantly due to socioeconomic and societal factors. | • Alcohol: Oral consumption. • Cigarettes: Inhalation. • Tramadol: Oral consumption (tablets or capsules). • Codeine-containing syrup: Oral consumption. • Cannabis: Inhalation (smoking) or oral consumption (edibles). | • The structured questionnaire was used. • Awareness of abused substances: The 21 questions in this section gauged respondent knowledge of the types and identification of drugs, their cost, where they are sold, their effect and the perceived motive for abuse of these drugs and substances. • Prevalence-related risk factors associated with abused substances (Section 3): This section included 16 questions designed to identify whether respondents were in an environment that could prompt them to abuse drugs and substances. • Reliability of the study questionnaire. A pilot test was conducted to verify the usability and appropriateness of the instrument | Low |
| 20. | Soremekun et al., (2020) [31] | Nigeria | Cross-sectional study | 1048 | Students | 12.5 years | Alcohol Tobacco Cannabis Cocaine Inhaled Substances Tranquilizers Sedatives Heroin Pharmaceutical opioids (codeine and tramadol) | Peer influence, Lack of parental supervision, Personality problems, need for energy to work long hours, Availability of drugs, Exposure to social media, Withdrawal symptoms, Purchasing power, Cultism | • Inhalation: Some students reported inhaling substances like perfume, gum, ammonia, spirit, and spray glue. • Oral Consumption: Alcohol and pharmaceutical opioids (like tramadol and codeine). • Inhalation: Tobacco and cannabis. | A pilot study was carried out using 10 students in Ikotun or Igando LCDA to pre-test the questionnaire, following which some modifications were made to simplify some words to match the students' reading level. | Low |

(Continued)

**Table 2.** (Continued)

| S/No | Author's Name/Year | Country | Study Type | Sample Size | Study Populations | Mean Age (Years) | Types of substances used | Factors influencing abuse | Route of substance use administration | Information on Study Tools | Quality Assessment |
|---|---|---|---|---|---|---|---|---|---|---|---|
| 21. | Umar et al., (2017) [32] | Nigeria | Cross-sectional study | 233 | Individuals in drug rehabilitation | 26.31 years | Cigarettes, alcohol, codeine, cannabis, benzodiazepine, heroin, tramadol, | - | • Cigarettes: Smoked. • Alcohol: Ingested orally (drinking) • Codeine: Taken orally as a syrup • Cannabis: Smoked, vaporized, or ingested orally • Benzodiazepine: Taken orally • Heroin: Injected, snorted, or smoked. • Tramadol: Taken orally | • Sociodemographic and Lifestyle Questionnaire • Adult ADHD ASRS (Internal consistency reliability of the continuous ASRS Screener was in the range 0.63–0.72) • FTND (The test–retest reliability is 0.75–0.91; Internal consistency was put at 0.65 and 0.69 for male and female) • DIVA • AAQoL • SSADDA | Low |
| 22. | Vigna-Taglianti et al., (2019) [33] | Nigeria | Cross-sectional study | 4078 | Students | 15.8 years | Tobacco, alcohol, cannabis, and other drugs | Poor social support and stigma | • Tobacco: Usually smoked in cigarettes or cigars. • Alcohol: Consumed as beverages like beer, wine, or spirits. • Cannabis: Often smoked in joints or pipes but can also be ingested in edibles. | • The questionnaire was a shortened version of that used in the EU-Dap study (www.eudap.net). • A pilot study was conducted in two classes in a school in Abuja in May 2015. | Low |

AAQoL—Adult ADHD Quality of Life

ADHD (ASRS)—Assessment of Attention-Deficit/Hyperactivity Disorder Adult Self-Report Scale [ASRS]

ASSIST—Alcohol, smoking and substance involvement screening test

CIDI—Composite International Diagnostic Interview

DIVA—Diagnostic Interview for ADHD in Adults

DUDIT—Drug Use Disorders Identification Test

FTND—Fagerstrom Test for Nicotine Dependence

SRQ-20—Self -Report Questionnaire

SSADDA—Semi-Structured Assessment for Drug Dependence and Alcoholism

YRBSS—Youth Risk Behaviour Surveillance System

relevant study characteristics from each included study. We employed a weighted approach to calculate the pooled prevalence, giving more weight to studies with larger sample sizes and more precise estimates. A random-effects model was used to account for the high heterogeneity anticipated across the studies, which was assessed using Cochrane's Q-test and the inconsistency index ($I^2$), with $I^2$ values $\geq 50\%$ indicating significant heterogeneity. We employed a random-effects model in the meta-analysis to account for the substantial heterogeneity ($I^2$ = 98–99%) observed across studies. This approach is appropriate given the variability in populations, settings, and substance use patterns, allowing for the generalization of findings beyond the included studies and better reflecting the true variation across different contexts in West Africa. To ensure the robustness of the pooled prevalence estimates, we performed a leave-one-out sensitivity analysis, systematically excluding each study from the analysis to determine its influence on the overall results. This step helped identify any studies that might disproportionately affect the findings. We also assessed publication bias by generating funnel plots for each substance, looking for symmetry as an indicator of low bias.

All analyses were carried out using R studio software (version 4.3.3), with the 'meta'and 'metafor'packages employed for conducting the meta-analyses and generating the necessary plots. The results were reported as percentages with 95% confidence intervals (CIs), providing clear and interpretable measures of substance use prevalence across the region.

### Quality and risk of bias assessment

An adapted version of the risk of bias tool for prevalence studies developed by Hoy and colleagues [21] was employed to assess the quality and risk of bias for studies that met the inclusion criteria. Each study underwent scrutiny across nine domains of bias, namely: description of the target population of the study, the sampling frame, sampling techniques, response rate, non-proxy collection of data, case definition of study, validity, and reliability of study instrument, mode of data collection, and appropriate description of numerator and denominator for the parameter of interest. The total score ranged from 0 to 9, with the overall score categorized as follows: 0–3: "low risk," 4–6: "moderate risk," and 7–9: "high risk" of bias. The assessment was carried out by three independent authors (FTA, ORA, OCE), and discrepancies were resolved by a fourth author (GOE). The details of the quality and risk of bias assessment is in S6 File.

## Results

### Selection of studies

Fig 1 illustrates the systematic study selection process. An initial search of four databases (PubMed, Web of Science, CINAHL, SCOPUS) and Google Scholar yielded 683 articles, spanning the period from 2005 to 2024. After 83 duplicates were removed, 600 articles underwent title and abstract screening. After applying the study exclusion criteria, 568 articles were excluded, leaving 32 for full-text evaluation. Subsequent thorough full-text screening resulted in the final inclusion of 22 articles [13, 22–42], which met the predefined inclusion criteria.

### Characteristics of the included studies

Table 2 shows synthesized data from 22 studies, encompassing 43,145 participants. The methodological composition of the included studies predominantly featured cross-sectional designs in 21 studies (95.5%) [13, 22–26, 28–42], with only one study (4.5%) adopting a case-control methodology [27] (See Table 2). Notably, the majority of the studies were carried out in Nigeria (18 studies) [13, 22–25, 29–34, 37–42], three studies in Ghana [26, 35, 36], and a singular study in a multi-country across Sub-Saharan Africa [28]. The age range across the included

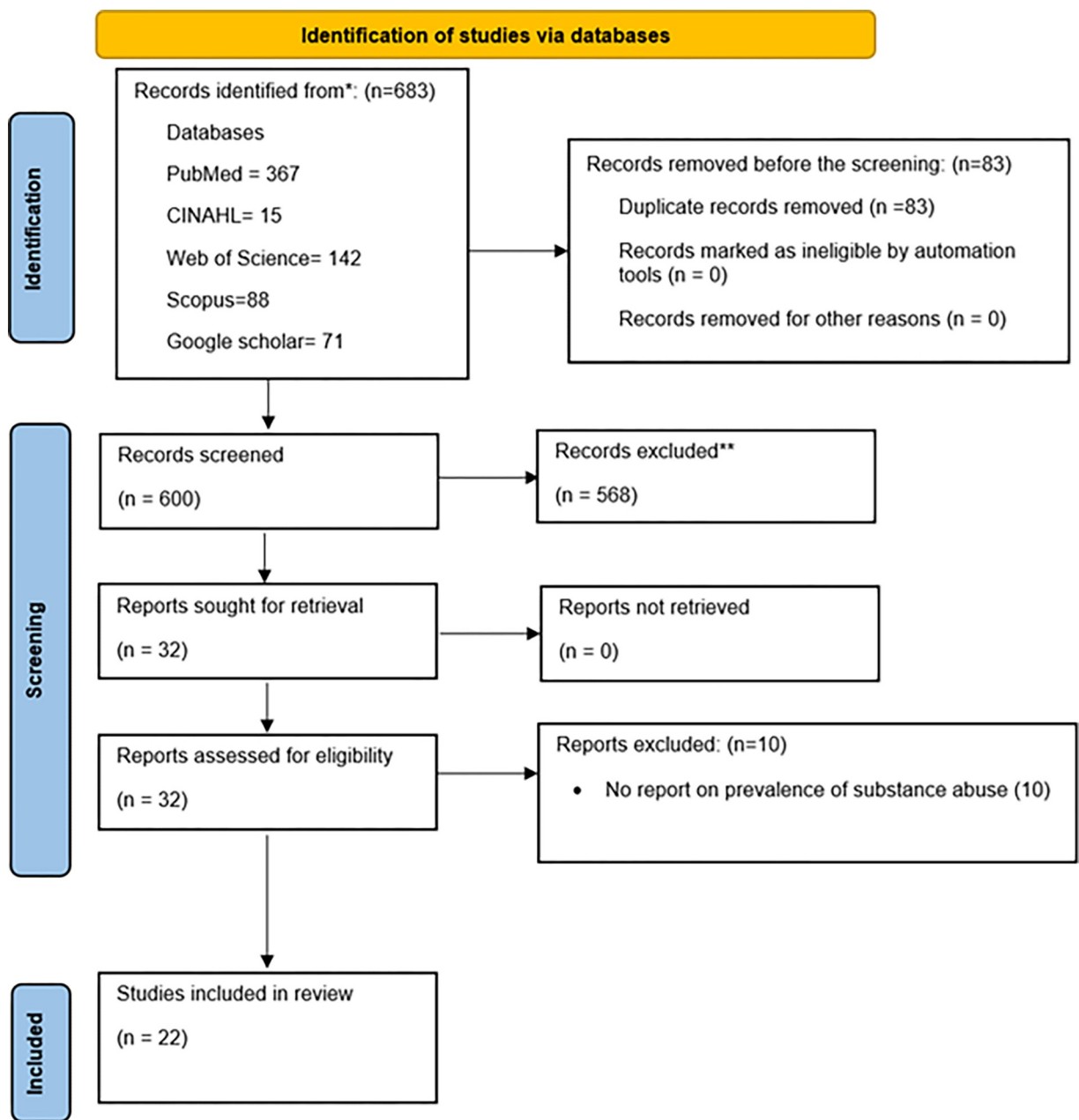

**Fig 1. Study Flowchart showing the flow of studies from retrieval to the final included studies.**

studies spans from adolescence (14 years) to older adulthood (60+ years). Additionally, a noteworthy proportion of the studies 11 studies (50%) [22, 27–31, 33, 34, 38, 41, 42] centered on student populations, two studies (9%) used internally displaced persons [24, 25], and a study used pregnant women as their primary participants [23].

## Types of substance used

Table 3 shows a broad spectrum of addictive substances used in West Africa. The substances reported span across various categories, including stimulants, prescription medications, illegal drugs, inhalants, and unconventional substances.

**Table 3. Categories of substance use.**

| Categories of Substance Used | Substance use |
|---|---|
| Stimulants and Natural Substances | Kolanut, coffee, green tea, and bitter kola |
| Commonly abused substances | Alcohol and cigarettes/nicotine |
| Prescription and over-the-counter medications | Tramadol, codeine, sedatives, sleeping pills and ephedrine |
| Illegal drugs | Cannabis/Marijuana, cocaine, heroin, Amphetamines, Methamphetamines, opiates/oxycodone |
| Inhalants and Unconventional Substances | Sniffing Agents, shoe polish, glue, pit toilet gas, lizard dung |
| Synthetic and designer drugs | Synthetic cannabinoids, Benzodiazepines, fentanyl |

From Table 3, kola nut and coffee were consistently reported as common stimulants, reflecting cultural practices. Bitter kola and green tea were also noted for their use as mild stimulants. Alcohol emerged as the most frequently abused substance, followed closely by nicotine in the form of cigarettes. These findings highlight the widespread acceptance and availability of these substances. Alcohol was identified as the most commonly abused substance, with nicotine, primarily in the form of cigarettes. These findings highlight the widespread acceptance and availability of these substances. Cannabis/marijuana was the most frequently reported illegal drug, followed by cocaine and heroin. The use of amphetamines and methamphetamines was also significant, indicating a trend toward the abuse of powerful stimulants. The use of inhalants like shoe polish, glue, and pit toilet gas highlights a desperate and dangerous form of substance use, particularly among marginalized populations. The mention of lizard dung as a hallucinogen underscores the extreme lengths to which individuals may go to achieve a high. Anabolic steroids were reported, suggesting a subset of the population using these substances for muscle growth and athletic performance despite their risks. The presence of synthetic cannabinoids such as THC and K2, as well as potent synthetic opioids like fentanyl, reflects the global trend of increasing synthetic substance use. Aphrodisiacs and various pharmaceutical opioids were also reported, indicating their non-medical use for purported enhancement effects.

## Routes of drug administration

From Table 2, oral Consumption is the most common route for ingesting substances such as alcohol, kolanut, bitter kola, green tea, and several pharmaceuticals like tramadol, sedatives, stimulants, benzodiazepines, codeine-containing syrups, and anabolic steroids. Substances like lizard dung and aphrodisiacs are also ingested orally. Oral consumption is preferred for ease of use, steady absorption, and prolonged effects.

Substances like cigarettes, cannabis, cocaine, heroin, synthetic cannabinoids, and solvents such as glue and shoe polish are predominantly smoked or inhaled. Inhalation provides a rapid onset of effects, making it a favored method for drugs like tobacco, marijuana, and volatile substances, which are commonly smoked or inhaled by users seeking immediate effects.

Cocaine and amphetamines are frequently snorted, allowing for quick absorption through the nasal mucosa. This method is often chosen for its rapid impact, though it may cause nasal irritation.

Drugs such as heroin, anabolic steroids, methamphetamine, and benzodiazepines are often injected, providing direct access to the bloodstream and resulting in immediate and intense effects. Injection is typically used for substances with a desired rapid onset and high bioavailability.

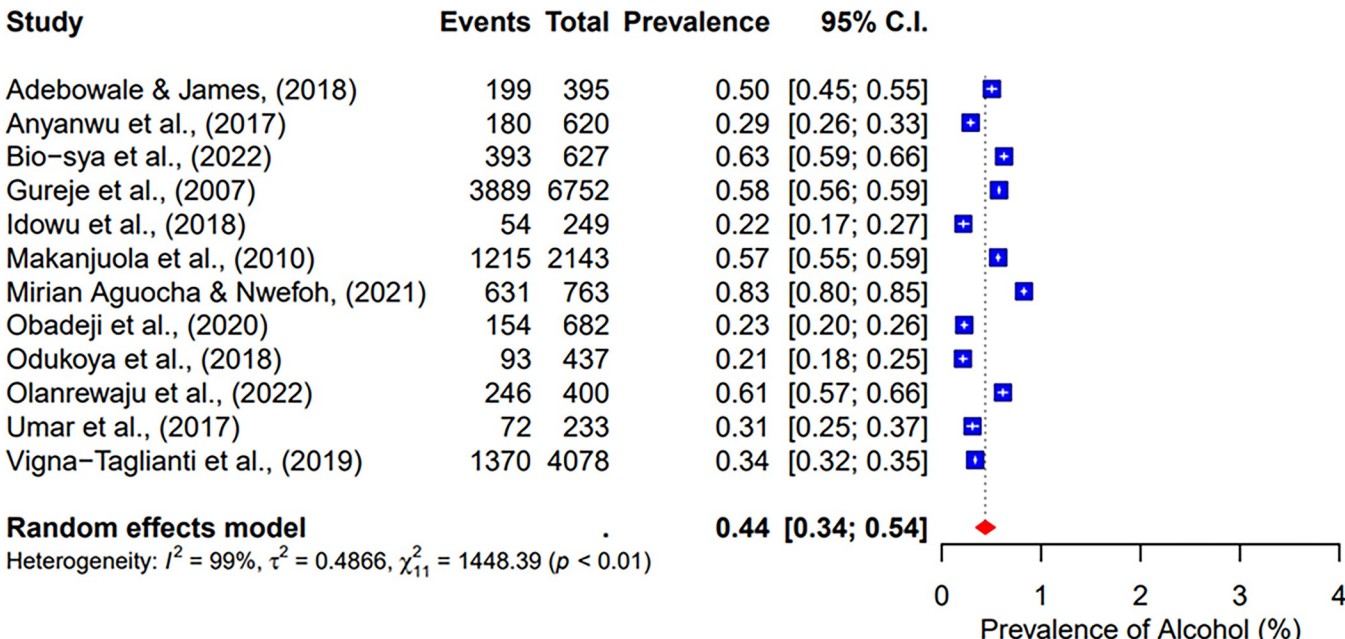

**Fig 2. Pooled prevalence of alcohol use in West Africa.**

Less common methods include transdermal absorption (as with fentanyl patches) and the brewing and ingestion of beverages like green tea, which are consumed for their psychoactive or stimulant properties.

## Pooled prevalence of substance use

Meta-analysis was conducted for substance use that had reported prevalence of at least three studies, such as alcohol, cannabis, cigarettes, cocaine, codeine, heroin, kolanut, marijuana, sedatives, tobacco, and tramadol. Substantial heterogeneity was observed across the included studies, indicating considerable variability in the effects reported ($I^2$ = 98–99%). Some studies reported the prevalence of benzodiazepines [32, 36] and amphetamines [22, 28], but they were not included in the meta-analysis due to the number of studies recorded.

Twelve included studies focused on alcohol use [23, 27, 29, 30, 32–34, 37, 38, 40–42]. This meta-analysis revealed a pooled prevalence of 44% [95% CI; 34–54%] for alcohol (See Fig 2).

The pooled prevalence of the ten studies that reported on cannabis abuse was 6% (95% CI; 3–10%) (See Fig 3). 5 studies indicated a pooled prevalence of 10% (95% CI; 3–31%) for cigarette use (See Fig 4). The pooled prevalence for cocaine, based on nine included studies, was 2% (95% CI; 1–7%) (See Fig 5). Additionally, tramadol was reported in 9 studies with a pooled prevalence of 30% (95% CI; 17–49%) (See Fig 6).

Furthermore, the pooled prevalence for codeine was 11% across six studies, heroin was 2% (95% CI; 0–10%), kolanut was 39% (95% CI; 25–55%), marijuana was 6% (95% CI; 2–19%), sedatives were reported at 13% (95% CI; 7–23%), and tobacco had a pooled prevalence of 9% (95% CI; 5–16%) (See Figs 7–12).

## Factors influencing substance use in West Africa

Analyzing the factors influencing substance use in West Africa reveals a complex array of determinants that span individual characteristics, social influences, socioeconomic factors,

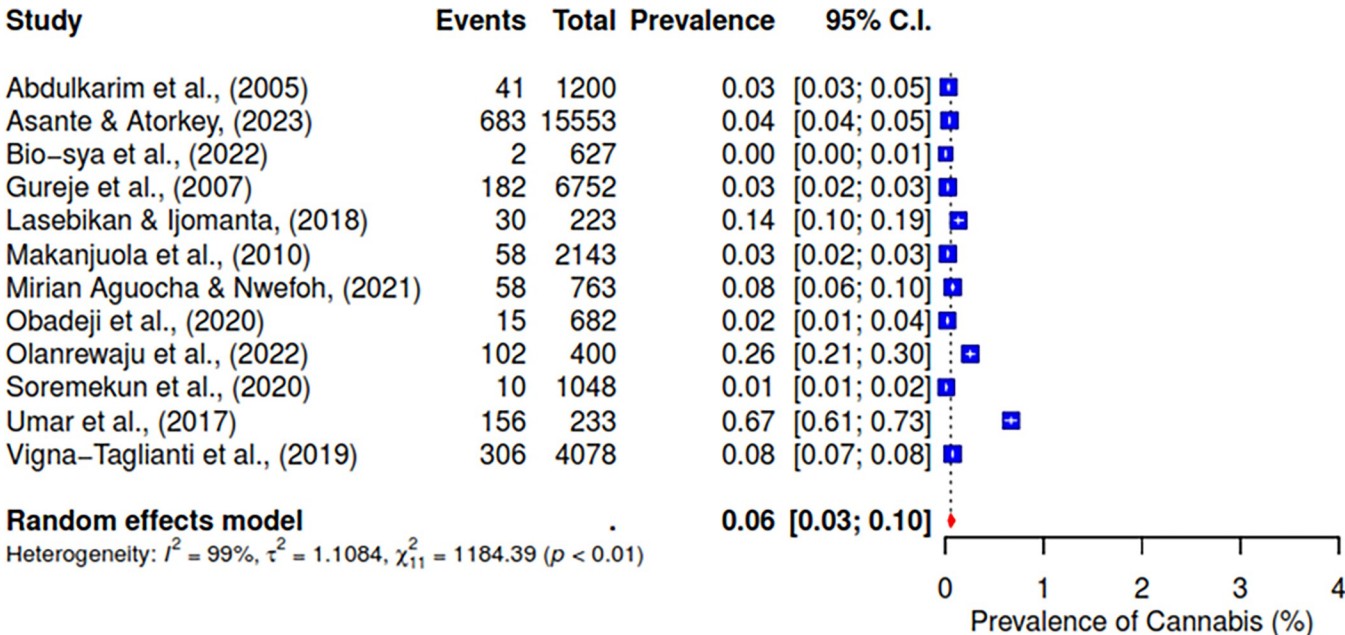

**Fig 3. Pooled prevalence of cannabis use in West Africa.**

and broader environmental conditions. These factors are categorized into key themes, each contributing to the region's prevalence and patterns of substance use (See Table 2).

Peer influence emerged as a significant factor across multiple studies [22, 31, 35, 38, 41]. A substantial proportion of individuals reported that peers heavily influenced their substance use initiation. The influence of peers was found to be more pronounced in urban settings, where social networks are larger and more diverse.

Family-related factors were also frequently cited as influential in substance use patterns [22, 26, 29, 31, 41, 42]. A lack of parental supervision and weak family control were consistently associated with higher rates of substance use. Studies highlighted that individuals from families with less parental monitoring or where parents themselves used substances were more likely to engage in substance use. Conversely, strong parental supervision and involvement were protective against substance use initiation.

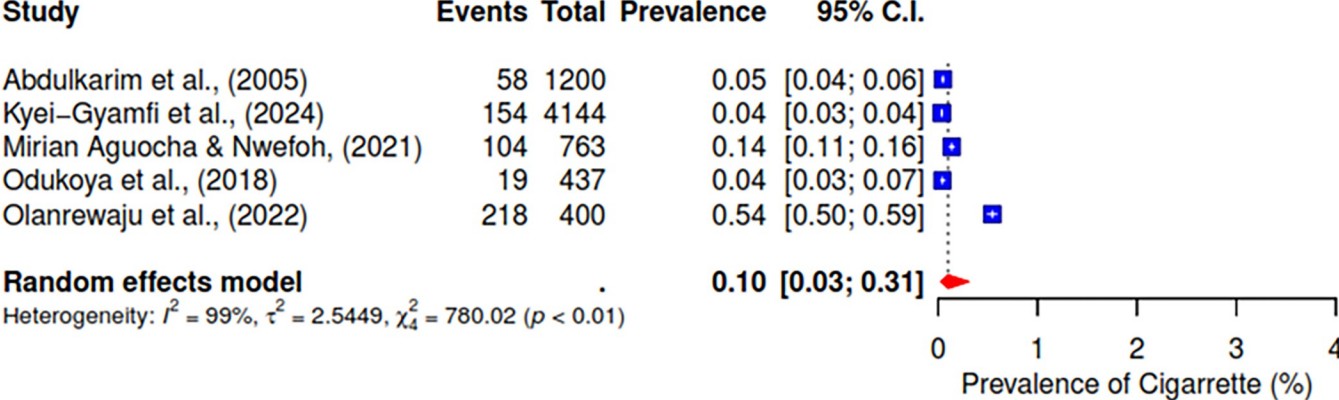

**Fig 4. Pooled prevalence of cigarette use in West Africa.**

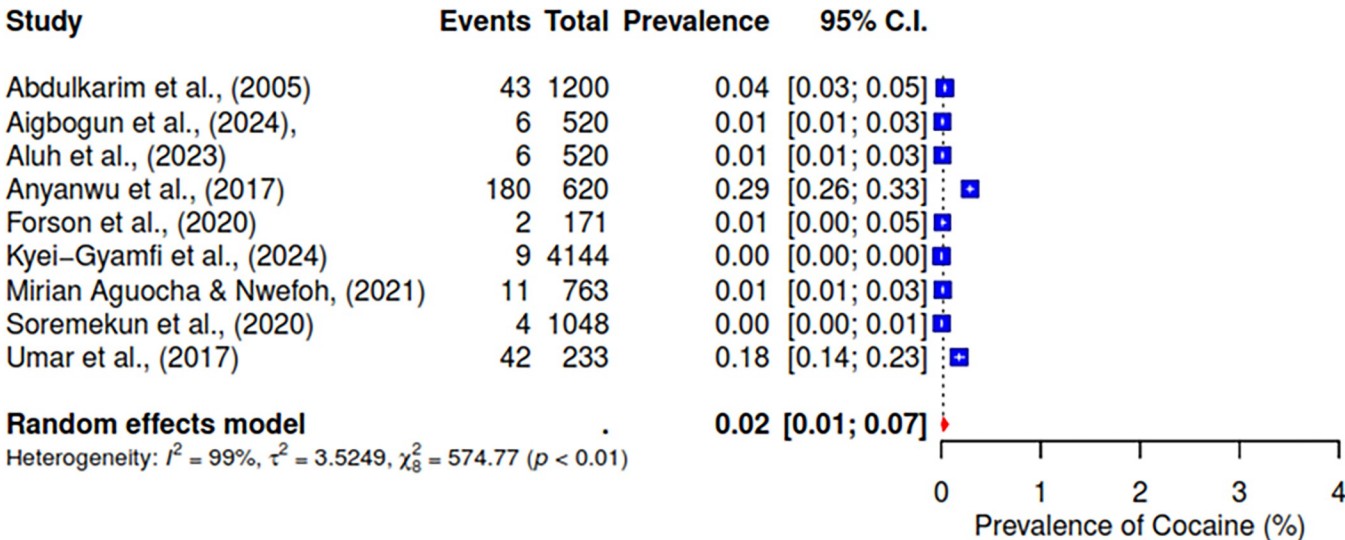

**Fig 5. Pooled prevalence of cocaine use in West Africa.**

Religious affiliation and the degree of religiosity were found to have a mixed impact on substance use [22, 37, 41]. In some studies, higher levels of religiosity were associated with lower rates of substance use, suggesting that religious practices and beliefs may act as protective factors.

Socioeconomic status, including income, employment, and education levels, influenced substance use [25–28, 37]. Individuals with lower income and educational attainment were more likely to engage in substance use, with unemployment being a significant risk factor. Substance use was also reported to be higher among those with past disciplinary actions at work, highlighting the role of occupational stress and instability in substance use behaviors.

The demographic analysis revealed that younger age groups, particularly adolescents and young adults, are at higher risk for substance use [26–28, 34, 37, 41]. Males were consistently

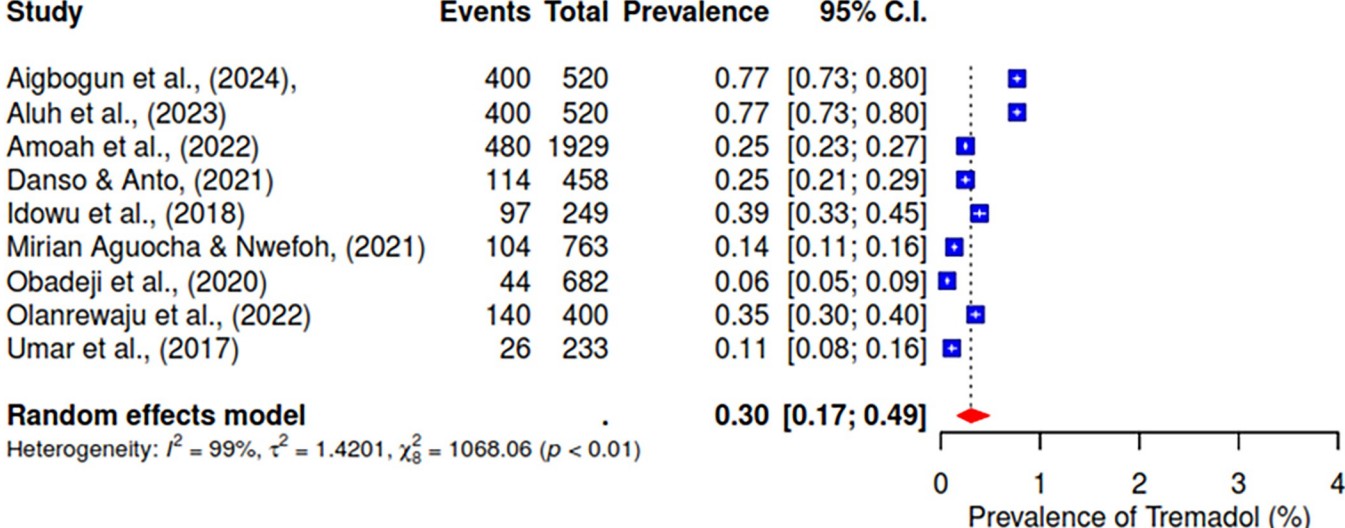

**Fig 6. Pooled prevalence of tramadol use in West Africa.**

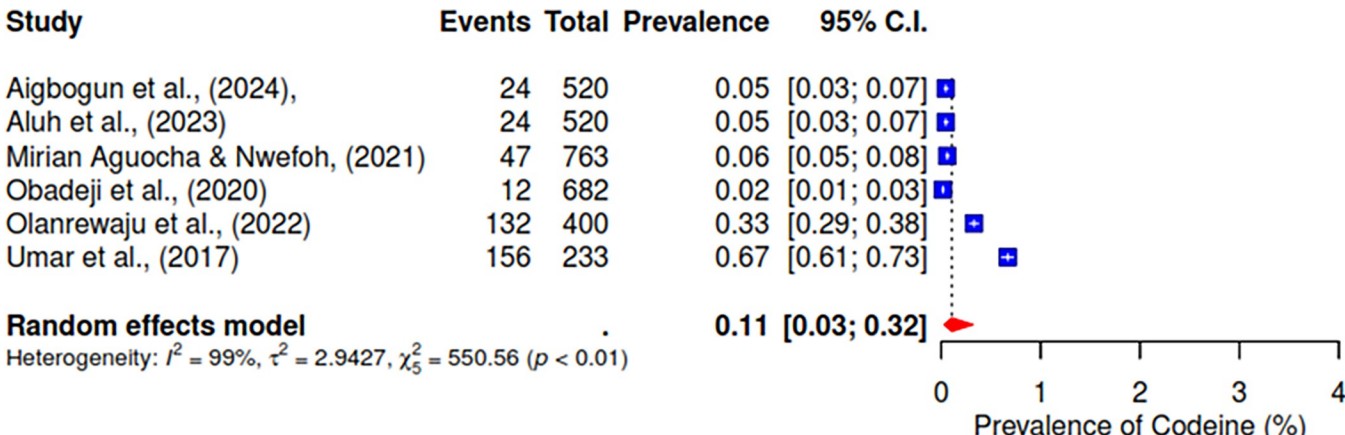

**Fig 7. Pooled prevalence of codeine use in West Africa.**

found to have higher rates of substance use compared to females. The data also indicated that socioeconomic class influences substance use, with those from lower-income backgrounds reporting higher use rates. Place of residence was another significant factor, with urban dwellers more likely to use substances than those in rural areas.

## Sensitivity analysis

A sensitivity analysis was conducted to assess the robustness of the overall prevalence estimate by using a leave-one-out approach. This method systematically excluded each study from the meta-analysis to evaluate its influence on the overall pooled prevalence of substance use in West Africa. This analysis aimed to determine whether any single study had a disproportionate impact on the overall results, which could indicate potential biases or outliers that might distort the findings.

The sensitivity analysis results showed that excluding any single study did not significantly alter the overall summary prevalence. The changes in the pooled prevalence estimates were minimal, with variations well within acceptable limits, indicating that the overall prevalence is stable and not unduly influenced by any particular study, thereby confirming the reliability and consistency of the findings across the included studies.

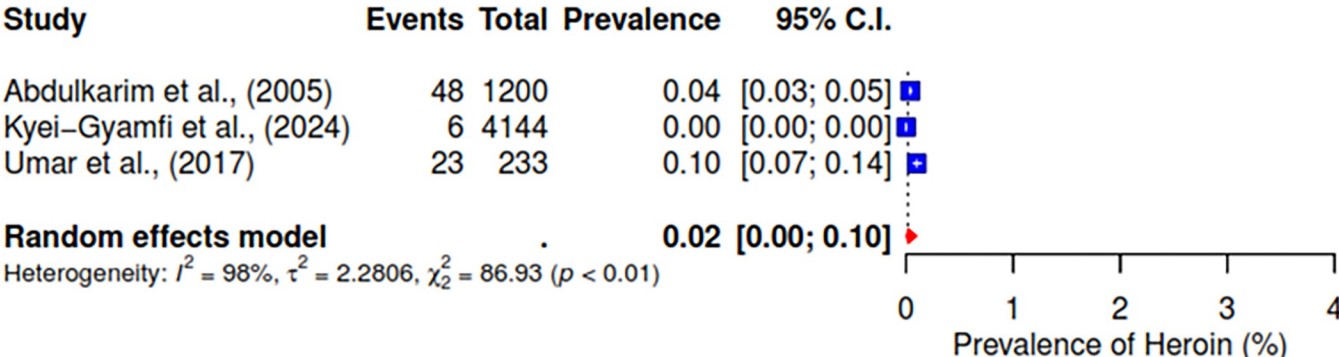

**Fig 8. Pooled prevalence of heroin use in West Africa.**

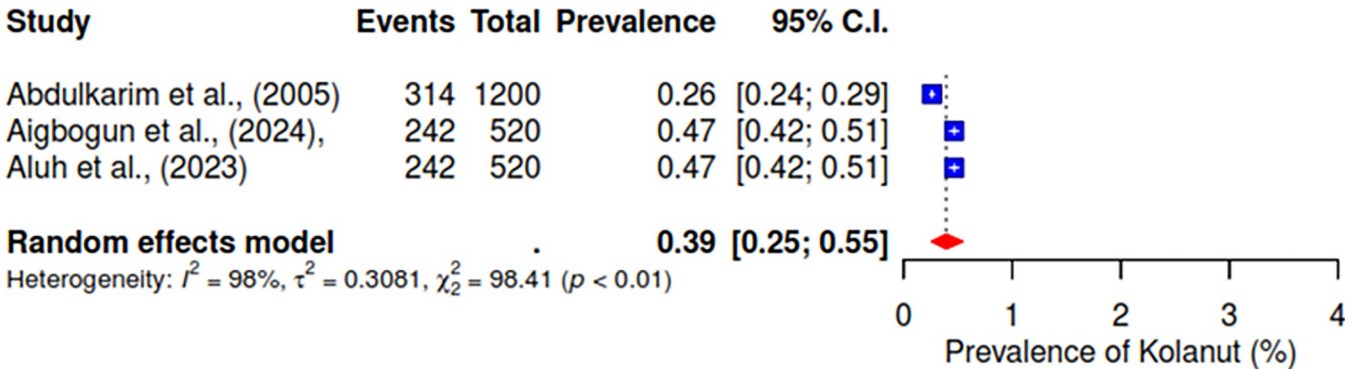

**Fig 9. Pooled prevalence of kolanut use in West Africa.**

The detailed results of the leave-one-out sensitivity analysis are presented in S7 File, which includes a breakdown of the pooled prevalence estimates with each study omitted. These findings support the robustness of the overall prevalence estimate and suggest that the conclusions drawn from this meta-analysis are reliable and generalizable across the studies included in the review.

## Quality and risk of bias assessment

Out of the 22 studies assessed, 95% (21/22) were found to have a low risk of bias, while one study was classified as having a moderate risk of bias. No studies were identified with a high risk of bias. This suggests that the majority of the studies included in this review were of high methodological quality, reducing the likelihood of bias in the reported findings as presented in S6 File.

## Publication bias

To assess potential publication bias in the meta-analysis, funnel plots were generated for each of the substances analyzed, including alcohol, cannabis, cigarettes, cocaine, codeine, kolanut, marijuana, sedatives, tobacco, and tramadol. The funnel plots for most substances displayed a relatively symmetrical distribution of study effects around the pooled prevalence estimate,

| Study | Events | Total | Prevalence | 95% C.I. |
|---|---|---|---|---|
| Aigbogun et al., (2024), | 31 | 520 | 0.06 | [0.04; 0.08] |
| Aluh et al., (2023) | 31 | 520 | 0.06 | [0.04; 0.08] |
| Forson et al., (2020) | 29 | 171 | 0.17 | [0.12; 0.23] |
| Kyei–Gyamfi et al., (2024) | 28 | 4144 | 0.01 | [0.00; 0.01] |
| Odukoya et al., (2018) | 86 | 437 | 0.20 | [0.16; 0.24] |
| **Random effects model** | | . | **0.06** | **[0.02; 0.19]** |

Heterogeneity: $I^2 = 99\%$, $\tau^2 = 2.0207$, $\chi^2_4 = 278.04$ ($p < 0.01$)

Prevalence of Marijuana (%)

**Fig 10. Pooled prevalence of marijuana use in West Africa.**

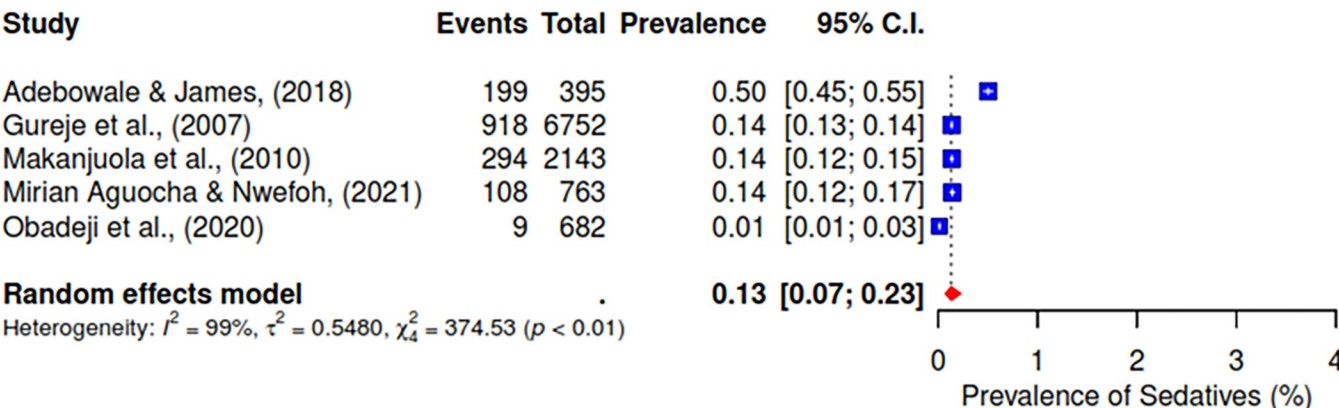

**Fig 11. Pooled prevalence of sedatives use in West Africa.**

suggesting a low risk of publication bias. However, slight asymmetry was observed in the funnel plots for substances like cannabis and cocaine, indicating the possibility of publication bias or the presence of heterogeneity among the studies included. Overall, while some minor indications of publication bias were noted, most of the funnel plots suggest that the findings of this meta-analysis are reliable and not significantly influenced by publication bias (See S8 File).

## Discussion

This systematic review and meta-analysis provide a comprehensive overview of the prevalence and patterns of drug abuse in West Africa, with a particular focus on Nigeria and Ghana. Our findings indicate a significant burden of substance use, with varying prevalence rates across different substances and demographic groups. The high heterogeneity observed in the studies underscores the complexity of substance use patterns in this region.

Our meta-analysis revealed a pooled prevalence of alcohol use at 44%, which is consistent with previous studies in Sub-Saharan Africa (SSA). For instance, Adeloye et al. (2019) reported a prevalence of 34.3% for alcohol use among Nigerian adults, reflecting the cultural acceptance and widespread availability of alcohol in the region in a systematic review. A systematic review that aimed to summarize the population level of alcohol use in SSA reported that the

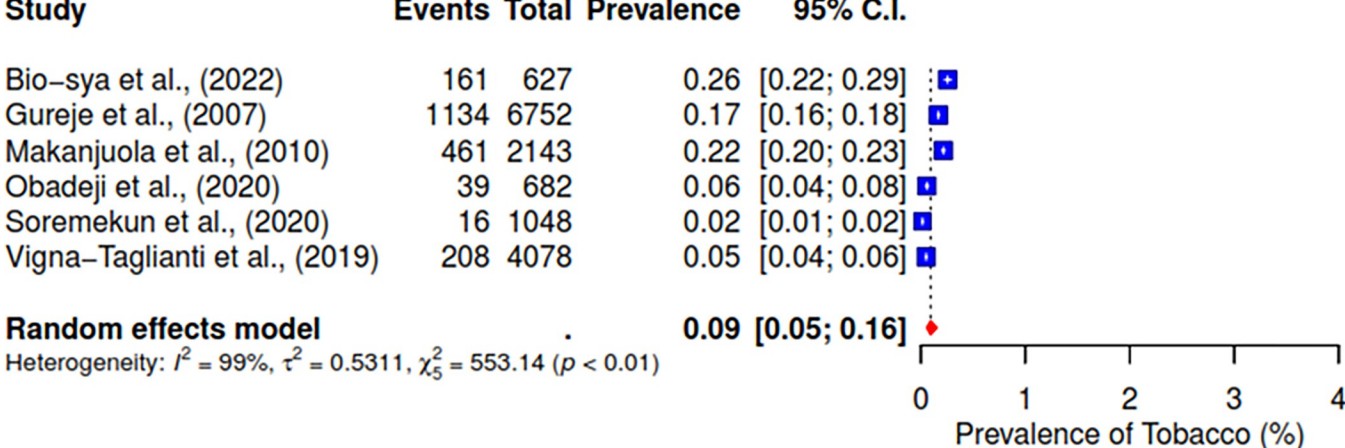

**Fig 12. Pooled prevalence of tobacco use in West Africa.**

prevalence of alcohol use among adolescents and adults is 23.3% and 34.9%, respectively [43]. Also, in a systematic review conducted in East Africa, the median prevalence of alcohol use among young people was recorded at 52% [44]. The high prevalence rates suggest an urgent need for comprehensive alcohol control policies, including public education campaigns, taxation, and restrictions on advertising and sales [45]. These strategies have been effective in other regions and could reduce alcohol use in West Africa [45, 46].

The pooled prevalence of cannabis use was found to be 6%. This result aligns with the findings of the United Nations Office on Drugs and Crime (UNODC), which reports cannabis as the most widely used illicit drug in Africa [12]. A study by Odek-Ogunde et al. (2004) in Kenya found a significant prevalence of cannabis use among youth, similar to our findings [47]. The consistent use of cannabis across various demographics highlights the need for targeted interventions that address the specific socio-cultural factors driving its use.

Our analysis identified a pooled prevalence of 10% for cigarette use. This finding is supported by the Global Adult Tobacco Survey (GATS) conducted in Nigeria, which reported a prevalence of 10.1% for current tobacco smoking among adults [48]. The high prevalence of nicotine use underscores the necessity for robust tobacco control measures, including smoking cessation programs and stricter enforcement of tobacco regulations. Comparative studies in South Africa, such as Peltzer et al. (2011), have shown similar trends, emphasizing the widespread challenge of tobacco use in the region [49].

The pooled prevalence for cocaine and heroin was found to be 2% each. These rates are consistent with global trends, where these substances are less prevalent compared to cannabis but pose significant public health risks [50, 51]. The lower prevalence in our study could be attributed to limited availability and higher costs compared to other substances. Similar studies in SSA have reported comparable findings, indicating the need for continued monitoring and targeted interventions to prevent an increase in their use [8, 52].

A significant finding in our study is the high prevalence of tramadol (30%) and codeine (11%) use. Tramadol abuse has been reported as a major public health issue in Nigeria, with regulatory interventions being implemented to curb its misuse. Several studies have highlighted the widespread non-medical use of tramadol, leading to significant health problems [53–55]. The abuse of these prescription drugs reflects a broader regional issue that requires targeted interventions, including stricter regulations on the sale and distribution of these substances and public education on their risks [52, 53, 56].

Kolanut use was prevalent at 39%, reflecting its cultural significance and stimulant properties. This finding is unique to West Africa, where kolanut is traditionally used for its stimulant effects [57, 58]. The high prevalence suggests that public health strategies should include culturally sensitive approaches to address its use, recognizing its traditional value while educating the population on its potential health risks [59].

The presence of synthetic cannabinoids and synthetic opioids like fentanyl indicates a global trend of increasing synthetic substance use [60]. The use of unconventional substances such as lizard dung and inhalants like shoe polish and glue among marginalized populations underscores the extreme measures individuals resort to for substance use [61]. These findings are corroborated by anecdotal evidence and smaller-scale studies reporting similar patterns [60]. Targeted interventions addressing the underlying social and economic determinants of health are essential to mitigate these risks.

In addition to tramadol and codeine, our study found a notable prevalence of other opioids, including heroin and synthetic opioids like fentanyl [31, 34, 36, 62]. The misuse of opioids is a growing concern globally, with the UNODC reporting significant increases in opioid abuse in various regions, including Africa [63]. The high prevalence of opioid misuse in our study underscores the need for robust opioid management programs, including prescription

monitoring, public education on the dangers of opioid abuse, and the availability of treatment options for those struggling with use.

The review also identified a significant prevalence of benzodiazepine use, with sedatives being reported at a prevalence of 13%. Benzodiazepines are commonly prescribed for anxiety and insomnia but are often misused for their sedative effects [64, 65]. A study by Maroh et al. (2018) highlighted the misuse of benzodiazepines in Nigeria, particularly among young adults [66]. The misuse of these drugs can lead to dependence and severe withdrawal symptoms, necessitating the implementation of stricter prescription guidelines and the availability of treatment and support for those affected [65, 67]. A systematic review of the epidemiology of benzodiazepine shows that people with psychiatric symptoms and disorders also appear to be more vulnerable to benzodiazepine misuse [65].

Our analysis revealed significant use of amphetamines [22, 28] and methamphetamines [34, 35]. Amphetamines are powerful stimulants often used non-medically for their performance-enhancing and euphoric effects [68, 69]. The use of these substances is particularly concerning due to their high potential for abuse and severe health consequences. Similar trends have been observed in other regions, with the UNODC reporting a rise in amphetamine-type stimulant use globally [10, 63]. Addressing this issue requires targeted prevention and intervention strategies, including education on the risks of amphetamine use and the development of treatment programs for those addicted to these substances.

The high prevalence of substance use, particularly alcohol and tramadol, underscores the urgent need for targeted public health interventions in West Africa [38, 56, 70]. Current efforts to address substance use may be insufficient, particularly in addressing the socio-cultural factors contributing to high rates of alcohol and stimulant use. Public health strategies should prioritize comprehensive education and prevention programs addressing the cultural acceptance of substances like alcohol and kolanut [71–73]. Additionally, stronger regulatory frameworks are needed to control the availability of prescription drugs, such as tramadol and codeine, which are increasingly being misused.

The study highlights several pivotal factors influencing substance abuse. Peer influence is a significant determinant, particularly among adolescents, where social acceptance drives substance use [74, 75]. The availability of substances also plays a critical role, with easier access leading to higher usage rates [76, 77]. Role models, especially parents and siblings, significantly impact substance use patterns, with lack of parental supervision being a key risk factor [78, 79]. Chronic pain and mental health disorders often lead to self-medication with substances like tramadol and codeine [80]. Additionally, individuals seeking enjoyment, those with personality traits like impulsivity, and those needing energy for extended work hours are more prone to substance use [75]. Effective prevention and intervention strategies must address these multifaceted influences to mitigate drug abuse.

The study's strength lies in its robust methodology, including adherence to PRISMA guidelines, using a random-effects model to account for variability across studies and a thorough sensitivity analysis that confirms the stability of the findings. By including a diverse range of studies spanning over two decades and covering various populations, such as adolescents, adults, and vulnerable groups like internally displaced persons, the review captures a broad spectrum of substance use behaviors. Additionally, the focus on emerging substances, such as tramadol and synthetic cannabinoids, highlights current public health concerns and provides critical data to inform targeted interventions.

Despite its comprehensive approach, the study has several limitations that must be acknowledged. The geographic concentration of the included studies, with a predominance from Nigeria and Ghana, limits the generalizability of the findings to the broader West African region, where less research on substance use has been conducted. Additionally, the high

heterogeneity observed across the studies ($I^2$ = 98–99%) indicates significant variability in prevalence estimates, which may result from differences in study design, populations, and contextual factors, potentially affecting the consistency of the findings. Also, the variability in the study tools used across the included studies. The differences in the questionnaires employed, along with their varying reliability may explain some of the observed heterogeneity in the results. Furthermore, the reliance on cross-sectional data restricts the ability to assess changes in substance use patterns over time or establish causality. Lastly, the underreporting of certain substances, particularly those less commonly studied or newly emerging, may lead to an incomplete representation of the substance use landscape in the region.

Based on our findings, we recommend comprehensive public education campaigns to raise awareness about the risks of substance use, focusing on alcohol, nicotine, and emerging drugs like tramadol and synthetic cannabinoids. There is also a need to enforce stricter regulations on the sale and distribution of prescription drugs, alcohol, and tobacco including increasing taxes, restricting sales to minors, and regulating advertising. Most importantly, there is a need to engage communities in designing and implementing substance use prevention and treatment programs. Community-based approaches can be more effective in addressing different populations' specific needs and cultural contexts.

## Conclusion

The study highlights the widespread and diverse nature of substance use in West Africa, with significant implications for public health and policy. By addressing the socio-cultural and economic factors contributing to substance use and implementing robust regulatory frameworks, it is possible to mitigate the public health impact of substance use in the region. Further research and targeted interventions are essential to address the emerging trends and specific challenges identified in this study.

## Supporting information

**S1 File. PRISMA checklist.**
(PDF)

**S2 File. Search terms and strategy.**
(PDF)

**S3 File. Excluded studies with reasons.**
(PDF)

**S4 File. Excluded studies with reasons.**
(XLSX)

**S5 File. Extraction sheet and data analysis.**
(XLSX)

**S6 File. Quality and risk of bias assessment.**
(PDF)

**S7 File. Leave one out result.**
(PDF)

**S8 File. Funnel plots.**
(ZIP)

## Author Contributions

**Conceptualization:** Godwin Omokhagbo Emmanuel, Folahanmi Tomiwa Akinsolu, Oliver Chukwujekwu Ezechi.

**Data curation:** Godwin Omokhagbo Emmanuel, Folahanmi Tomiwa Akinsolu, Olunike Rebecca Abodunrin, Oliver Chukwujekwu Ezechi.

**Formal analysis:** Godwin Omokhagbo Emmanuel, Folahanmi Tomiwa Akinsolu, Olunike Rebecca Abodunrin, Oliver Chukwujekwu Ezechi.

**Investigation:** Folahanmi Tomiwa Akinsolu, Olunike Rebecca Abodunrin, Oliver Chukwujekwu Ezechi.

**Methodology:** Godwin Omokhagbo Emmanuel, Folahanmi Tomiwa Akinsolu, Olunike Rebecca Abodunrin, Oliver Chukwujekwu Ezechi.

**Project administration:** Godwin Omokhagbo Emmanuel.

**Resources:** Godwin Omokhagbo Emmanuel.

**Software:** Godwin Omokhagbo Emmanuel, Folahanmi Tomiwa Akinsolu, Olunike Rebecca Abodunrin.

**Supervision:** Godwin Omokhagbo Emmanuel, Folahanmi Tomiwa Akinsolu, Oliver Chukwujekwu Ezechi.

**Validation:** Godwin Omokhagbo Emmanuel, Olunike Rebecca Abodunrin.

**Visualization:** Godwin Omokhagbo Emmanuel.

**Writing – original draft:** Godwin Omokhagbo Emmanuel, Folahanmi Tomiwa Akinsolu, Olunike Rebecca Abodunrin, Oliver Chukwujekwu Ezechi.

**Writing – review & editing:** Godwin Omokhagbo Emmanuel, Folahanmi Tomiwa Akinsolu, Olunike Rebecca Abodunrin, Oliver Chukwujekwu Ezechi.

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
