## [Decision Letter · Decision Letter 0]

22 Sep 2024

PGPH-D-24-01873

Prevalence and Patterns of Substance Use in West Africa: A Systematic Review and Meta-analysis

Dear Dr. Akinsolu,

Thank you for submitting your manuscript to PLOS Global Public Health. After careful consideration, we feel that it has merit but does not fully meet PLOS Global Public Health’s publication criteria as it currently stands. Therefore, we invite you to submit a revised version of the manuscript that addresses the points raised during the review process.

We look forward to receiving your revised manuscript.

Kind regards,

Massimiliano Orri, PhD

Academic Editor

Journal Requirements:

1. As required by our policy on Data Availability, please ensure your manuscript or supplementary information includes the following: 

- https://bsdwebstorage.blob.core.windows.net/ejournals-2220-3206/WJPv12i10.pdf

In your revision ensure you cite all your sources (including your own works), and quote or rephrase any duplicated text outside the methods section. Further consideration is dependent on these concerns being addressed.

3. We have amended your Competing Interest statement to comply with journal style. We kindly ask that you double check the statement and let us know if anything is incorrect. 

4. In the online submission form, you indicated that "All datasets generated and analyzed, including the study protocol, search strategy, list of included and excluded studies, data extracted, analysis plans, and quality assessment, are available in the article and upon request from the corresponding author.". 

3. Uploaded as supplementary information.

5. Please provide separate figure files in .tif or .eps format.

Additional Editor Comments (if provided):

Reviewers' comments:

Reviewer's Responses to Questions

**Comments to the Author**

1. Does this manuscript meet PLOS Global Public Health’s publication criteria? Is the manuscript technically sound, and do the data support the conclusions? The manuscript must describe methodologically and ethically rigorous research with conclusions that are appropriately drawn based on the data presented.

Reviewer #1: Yes

Reviewer #2: Yes

Reviewer #3: Yes

2. Has the statistical analysis been performed appropriately and rigorously?

Reviewer #1: Yes

Reviewer #2: Yes

Reviewer #3: Yes

3. Have the authors made all data underlying the findings in their manuscript fully available (please refer to the Data Availability Statement at the start of the manuscript PDF file)?

Reviewer #1: Yes

Reviewer #2: Yes

Reviewer #3: Yes

4. Is the manuscript presented in an intelligible fashion and written in standard English?

Reviewer #1: Yes

Reviewer #2: Yes

Reviewer #3: Yes

5. Review Comments to the Author

Reviewer #1: High level Public Health relevant study. A few areas need to be addressed to improve the worker further before publication.

1. PICO or PICOS framework is designed and used for identifying studies that use experimental design (quasi or non-quasi research design). As stated, this meta-analysis considered observational studies known for not having intervention component. Hence, I would suggest removing the intervention (I) component from the framework.

2. Traditionally, any form of bias is required, as much as possible, to be prevented and not corrected. Funnel Plot and Egger’s test are carried out at analytical level to explore possibility of bias in meta-analysis. However, bias could be prevented before getting to analytical stage and this should be presented as necessary steps in the methodology. Hence, the following steps have been suggested to prevent publication bias right before getting into analytical stage

• Using registered reports.

• Comparing the results of published and unpublished papers on the same research topic.

• Consulting journals that publish negative findings, such as Positively Negative (PLOS One), Journal of Articles in Support of the Null Hypothesis, or Journal of Pharmaceutical Negative Results

Please use any or all of the three above steps (especially the third one) to show how you have prevented publication bias. Please, ensure that this is included in your PRISMA flow.

3.lease offer more discussion to show the rationale for using random effect, as opposed to fixed effect model.

4. There are questionnaires that each eligible study employed to obtain data. Please provide the information (name of the questionnaire, reliability score i.e. Cronbach alpha score) about each of these questionnaires and include all in table 2 and comment on all. This may be the explanatory source for the high heterogeneity observed in this study

Reviewer #2: Prevalence and Patterns of Substance Use in West Africa: A Systematic Review and Meta-analysis

Overall Assessment:

This systematic review and meta-analysis provides a comprehensive overview of substance use patterns and prevalence in West Africa, with a focus on Nigeria and Ghana. The study addresses an important public health issue and synthesizes data from 22 studies involving over 43,145 participants. The methodology is robust, following PRISMA guidelines and employing appropriate statistical techniques. The findings offer valuable insights into substance use trends in the region, highlighting high prevalence rates for alcohol, tramadol, and other substances. However, there are some limitations and areas for improvement that should be addressed.

Strengths:

1. Comprehensive search strategy across multiple databases

2. Adherence to PRISMA guidelines for systematic reviews

3. Rigorous data extraction and quality assessment process

4. Sensitivity analysis to evaluate the robustness of findings

5. Detailed analysis of substance types, prevalence rates, and influencing factors

6. Discussion contextualizes findings within existing literature and regional trends

Limitations:

1. High heterogeneity (I2 = 98-99%) across studies, which limits generalizability. If this is due to

2. Geographic concentration on Nigeria and Ghana, limiting representation of broader West Africa

3. Limited analysis of gender differences in substance use patterns

Suggestions for Improvement:

1. Provide more detail on the search strategy, including specific search terms used

2. The introduction provides good context but could benefit from a clearer statement of objectives

3. The results are comprehensive, but some of the figures could be improved for clarity, particularly the PRISMA diagram. Also, the tables should be labeled for easy correspondence while reading the paper.

Conclusion:

This systematic review and meta-analysis makes a valuable contribution to understanding substance use patterns in West Africa. With some revisions to address the limitations noted above, it would be suitable for publication. The findings have important implications for public health interventions and policy in the region.

Reviewer #3: Line 134: Studies Eligibility Criteria : Studies included individuals of all (should be 'ANY') age groups residing in West African countries. But many studies included in analysis has specific population( student, pregnant women, adolescents, etc). Suggested to change the criteria to ANY

Line 231: Alcohol emerged as the most frequently abused substance, followed closely by nicotine in the form of cigarettes. These findings highlight the widespread acceptance and availability of these substances.

It’s a repeated sentence. It is suggested a single table of pooled prevalence of various types of substance use.

Line 235: The use of amphetamines and methamphetamines was also significant, - I cannot see this in results tables or figures.

6. PLOS authors have the option to publish the peer review history of their article (what does this mean?). If published, this will include your full peer review and any attached files.

**Do you want your identity to be public for this peer review?** For information about this choice, including consent withdrawal, please see our Privacy Policy.

Reviewer #1: **Yes: **Adeniyi A. Adeboye

Reviewer #2: **Yes: **Queen Esther Adeyemo

Reviewer #3: No

---

## [Decision Letter · Decision Letter 1]

20 Nov 2024

Prevalence and Patterns of Substance Use in West Africa: A Systematic Review and Meta-analysis

PGPH-D-24-01873R1

Dear Dr. Akinsolu,

We are pleased to inform you that your manuscript 'Prevalence and Patterns of Substance Use in West Africa: A Systematic Review and Meta-analysis' has been provisionally accepted for publication in PLOS Global Public Health.

Best regards,

Massimiliano Orri, PhD

Academic Editor

Reviewer Comments (if any, and for reference):

Reviewer's Responses to Questions

**Comments to the Author**

1. If the authors have adequately addressed your comments raised in a previous round of review and you feel that this manuscript is now acceptable for publication, you may indicate that here to bypass the “Comments to the Author” section, enter your conflict of interest statement in the “Confidential to Editor” section, and submit your "Accept" recommendation.

Reviewer #1: All comments have been addressed

Reviewer #2: All comments have been addressed

Reviewer #3: All comments have been addressed

2. Does this manuscript meet PLOS Global Public Health’s publication criteria? Is the manuscript technically sound, and do the data support the conclusions? The manuscript must describe methodologically and ethically rigorous research with conclusions that are appropriately drawn based on the data presented.

Reviewer #1: Yes

Reviewer #2: Yes

Reviewer #3: Yes

3. Has the statistical analysis been performed appropriately and rigorously?

Reviewer #1: Yes

Reviewer #2: Yes

Reviewer #3: Yes

4. Have the authors made all data underlying the findings in their manuscript fully available (please refer to the Data Availability Statement at the start of the manuscript PDF file)?

Reviewer #1: Yes

Reviewer #2: Yes

Reviewer #3: Yes

5. Is the manuscript presented in an intelligible fashion and written in standard English?

Reviewer #1: Yes

Reviewer #2: Yes

Reviewer #3: Yes

6. Review Comments to the Author

Reviewer #1: (No Response)

Reviewer #2: This paper has been written with much rigor and clarity. It is an important study for understanding substance use and disorders in West Africa. Great work!

I only have some minor comments for the authors to address.

I think the authors are confusing substance use for substance use disorders, with special attention to the definition in line 65. This definition is wrong. Substance us is not a chronic problem. There is a clear difference between substance use and substance use disorders. Please address this.

In Table 2, the columns empty with the signs “–“mean what? If these information are not available, it should be clearly stated.

Line 249: remove the extra period after the cigarette.

Reviewer #3: (No Response)

7. PLOS authors have the option to publish the peer review history of their article (what does this mean?). If published, this will include your full peer review and any attached files.

**Do you want your identity to be public for this peer review?** For information about this choice, including consent withdrawal, please see our Privacy Policy.

Reviewer #1: **Yes: **Adeniyi Abolaji Adeboye

Reviewer #2: **Yes: **Queen E. Adeyemo

Reviewer #3: No
